# Biomarkers in AL Amyloidosis

**DOI:** 10.3390/ijms222010916

**Published:** 2021-10-09

**Authors:** Despina Fotiou, Foteini Theodorakakou, Efstathios Kastritis

**Affiliations:** Department of Clinical Therapeutics, School of Medicine, National and Kapodistrian University of Athens, 11528 Athens, Greece; desfotiou@gmail.com (D.F.); foteint@gmail.com (F.T.)

**Keywords:** AL amyloidosis, biomarker, Mayo cardiac staging, NT-proBNP, plasma cell clone, serum free light chains, minimal residual disease, growth differentiation factor-15

## Abstract

Systemic AL amyloidosis is a rare complex hematological disorder caused by clonal plasma cells which produce amyloidogenic immunoglobulins. Outcome and prognosis is the combinatory result of the extent and pattern of organ involvement secondary to amyloid fibril deposition and the biology and burden of the underlying plasma cell clone. Prognosis, as assessed by overall survival, and early outcomes is determined by degree of cardiac dysfunction and current staging systems are based on biomarkers that reflect the degree of cardiac damage. The risk of progression to end-stage renal disease requiring dialysis is assessed by renal staging systems. Longer-term survival and response to treatment is affected by markers of the underlying plasma cell clone; the genetic background of the clonal disease as evaluated by interphase fluorescence in situ hybridization in particular has predictive value and may guide treatment selection. Free light chain assessment forms the basis of hematological response criteria and minimal residual disease as assessed by sensitive methods is gradually being incorporated into clinical practice. However, sensitive biomarkers that could aid in the early diagnosis and that could reflect all aspects of organ damage and disease biology are needed and efforts to identify them are continuous.

## 1. Introduction

Immunoglobulin light chain (AL) amyloidosis is a rare and heterogenous hematological disorder characterized by the production and extracellular deposition of misfolded immunoglobulin free light chains (FLCs) that form amyloid fibrils and which originate from plasma cell or other B-cell clones [1]. FLC-derived amyloid fibrils deposit on target tissues causing disruption of their architecture and organ dysfunction [2]. Outcomes are currently poor and the non-specific and insidious nature of symptom presentation leads to delayed diagnosis and early mortality. Sensitive biomarkers for early diagnosis and biomarkers with prognostic value are necessary to guide appropriate treatment decisions and to serve as surrogate markers for clinical endpoints to allow new drug approvals by regulatory authorities. 

Prognosis in AL amyloidosis and particularly early mortality are largely dependent on the pattern and degree of end-organ-damage by the amyloidogenic FLC. Cardiac involvement is the critical determinant of survival and efforts have largely focused on devising prognostic tools that assess the degree of cardiac dysfunction [1,3,4]. The current standard staging systems include biomarkers of cardiac dysfunction—cardiac troponins, brain natriuretic peptide (BNP) and its N-terminal pro-brain natriuretic peptide (NT-proBNP) [5,6,7]. Factors that relate to the underlying clonal disorder also have prognostic value particularly for longer-term outcomes and treatment response. A revised version includes FLCs as a marker of the burden of circulating free light chains [8]. Renal involvement is a major cause of morbidity due to risk for progression to end-stage renal disease requiring dialysis but the search for renal biomarkers has always come second given the dramatic effect of cardiac involvement in overall mortality. A renal staging system has also been proposed and is widely used [9,10].

As early mortality remains a significant issue, prognostic factors in early survivors with outcomes based on landmark analysis are expected to differ from those used for “intention-to-treat analysis” at baseline. Current staging systems seem to fall short in their prognostic ability in the longer-term. 

In this paper we will review the currently used biomarkers in AL amyloidosis and the unmet needs. Given disease complexity, biomarkers have gained importance over recent years, as accurate estimates of prognosis and subsequent disease monitoring are necessary. A biomarker that entails all aspects of the disease in a sensitive but also specific manner (cardiac damage, renal failure, monoclonal plasma cell burden, and toxic free light chain circulation) is missing [11]. A number of new biomarkers have been reported in recent years and although none has made it to everyday clinical practice or has been incorporated in clinical trial design, emerging new data is promising. 

## 2. Current Prognostic Staging Systems and Biomarkers

Given the significant prognostic impact of cardiac involvement with early death it is only reasonable that markers of cardiac injury and dysfunction have emerged over the years as powerful prognostic factors. 

Serum levels of NT-proBNP and cardiac troponin T (cTnT) were first found to predict survival in several cohorts of AL patients [5,7,12,13]. They were later incorporated into the first widely used staging system for AL amyloidosis (Mayo 2004) which defines three stages [6]. The composition and biomarker thresholds were subsequently revised and two modifications of the original score are widely accepted [7,8] (Table 1). The original Mayo score includes NT-proBNP and troponin T and the 2012 revision incorporates the difference between the involved and the uninvolved FLC. The European version of the 2004 Mayo system identifies patients with very high NT-proBNP levels as having very poor outcomes and splits stage III into two stages (IIIa and IIIb) based on the 8500 ng/L cutoff for NT-proBNP. 

Cardiac troponins T (cTnT) and I (cTnI) are very sensitive and specific markers of cardiac injury. Sensitivity is increased with the use of next-generation high-sensitivity assays [13]. Subtle myocyte damage leads to release of troponin and abnormal serum levels due to high concentrations within the myocardium, a high release ratio, and prolonged elevation after injury [14,15,16]. 

Deposition of amyloid fibrils and direct toxicity of the clonal free light chains in the myocardium leads to increased wall stress, mostly in the left ventricle, the activation of p38-MAPkinase pathway, and induction of proBNP in the cardiac myocytes. Once released, proBNP, which is a 108-amino acid propeptide, is cleaved into the active brain natriuretic peptide (BNP) and a leader sequence, NT-proBNP. NT-proBNP has emerged as a very sensitive marker of cardiac failure which is elevated in the asymptomatic stages of left ventricular dysfunction. NT-proBNP levels change quickly with treatment and therefore may also serve as a reliable indicator of treatment response [17]. The FLC test, allows for sensitive quantification of the involved FLC and underlying clonal disorder [18,19,20], and the addition of dFLC on cardiac biomarkers in the Mayo 2012 revision added more long-term prognostic information, since FLC levels reflect the underlying clonal disease burden [21,22].

The combination of troponin T and NT-proBNP provides an objective, reproducible risk assessment tool based on biochemical criteria. However, serum troponins and NT-proBNP levels are influenced by renal dysfunction and other factors such as fluid overload and atrial arrythmias. Their reliability in these settings is therefore questioned [23,24] and regulatory authorities remain skeptical about their use as primary endpoints in clinical trials. Dittrich et al. [25] compared the performance of the staging systems in the context of atrial fibrillation and renal dysfunction (estimated glomerular filtration rate (eGFR) < 50 mL/min/1.73 m^2^). The least precise staging system in the entire cohort was Mayo2004 but European modification with addition of stage IIIb was the most robust. Performance of all systems was almost unaffected by renal impairment but less so by atrial arrythmias. 

While heart involvement and cardiac biomarkers have driven patients’ survival and therapeutic options, renal biomarkers are used in order to predict renal survival and progression to dialysis. 

The use of proteinuria and eGFR have been validated in thousands of patients with renal diseases and they have been used as endpoints in clinical trials. Current staging systems for prognosis of the renal disease and kidney response to treatment are based on serial eGFR and proteinuria assessments. However, neither marker is specific nor sensitive nor do they represent direct injury of kidney cells. Notably their levels are affected when kidney damage may already be advanced and irreversible, while heart failure can also contribute to impaired renal function (as measured by eGFR). In a small percentage of patients, where amyloid deposits affect renal vasculature rather than glomeruli, rapid deterioration of creatinine may be the dominant feature of renal involvement without proteinuria [26], so renal response criteria based on proteinuria cannot be reliable. Moreover, hematological response to therapy is assessed as early as in the first month after initiation of therapy. With the established criteria, renal response is time-dependent and a profound decrease in proteinuria may be observed after a period of 6 to 12 months following hematologic response. 

However, proteinuria with 24 h urine collection and eGFR are easy to measure and widely available. Palladini et al. in 2014 developed and validated a renal staging system defined by the cutoffs of 5 gr/24 h for proteinuria and 50 mL/min/1.73 m^2^ for eGFR. According to their results, the probability of progression to dialysis within 3 years was 0% to 4% for renal stage I, 7% to 30% for renal stage II, and 60% to 85% for renal stage III [9]. They also defined renal response and progression criteria—a reduction in eGFR from baseline of 25% or more was associated with higher probability of end-stage renal disease (ESRD) and defined “renal progression”, while a reduction in proteinuria of more than 30% from baseline or below 0.5 gr/24 h, without fulfilling criteria for renal progression, was correlated with longer renal survival [9] and defined as “renal response”. It was proposed that these criteria could be used in order to evaluate response to treatment along with hematologic response and cardiac response criteria as early as in 3 and 6 months after start of therapy. 

Even though creatinine, eGFR, and proteinuria are established biomarkers, they can be influenced by many factors such as hydration status, diuretic use, fluctuations in weight, and comorbidities. According to Palladini criteria, progression of renal disease is only defined by eGFR (that is serum creatinine); however, the identification of patients with inadequate renal response has clinical relevance. Because of the limitations with the use of eGFR alone, the effort to develop more sensitive biomarkers continues. In another approach, Kastritis et al. introduced the 24 h proteinuria to eGFR ratio (24 hUPr/eGFR) as a potentially more sensitive and accurate biomarker for evaluation of renal progression. The ratio incorporates both proteinuria and eGFR without additional testing. The analysis showed that an increase by at least 25% or a ratio >100 at 3 and 6 months after initiation of therapy correlated with higher risk of ESRD. In addition, patients with renal involvement could be stratified in three stages based on baseline 24 hUPr/eGFR ratio and these stages could also predict the risk of progression to dialysis in accordance to Palladini criteria. Thus, a ratio below 30 was associated with 0% risk of progression to dialysis at 3 years, a ratio between 30 and 99 was associated with 11% probability and a ratio over 100 with 46% probability of dialysis at 3 years [10]. This staging system could further separate patients at intermediate risk per the Palladini system, providing additional prognostic information. The two staging systems have been compared in an independent cohort, revealing their limitations [27]. 

## 3. Other Markers of Organ-Related Dysfunction Associated with Prognosis 

Table 2 summarizes markers of organ dysfunction that have been evaluated to have adverse outcomes in patients with AL amyloidosis over the past years. 

### 3.1. Markers of Cardiac Dysfunction 

In addition to NT-proBNP and cardiac troponin, low arterial blood pressure (<100 mmHg) [7], high New York Heart Association class (NYHA) (>2) [28], and the presence of atrial arrythmias [25,34] have been linked to worse prognosis in patients with AL amyloidosis. The presence of atrial arrythmia retained its prognostic adverse value in a multivariate model which also included age, dFLC, and cardiac staging systems in a recent study by Dittrich et al. [25] In the study by Sidana et al. in 2019, Holter monitoring was used for arrhythmia evaluation and atrial fibrillation (HR, 2.5; 95% CI, 1.2–5.0; *p* = 0.02) and non-sustained ventricular tachycardia (SVT) (HR, 2.0; 95% CI, 1.1–3.5; *p* = 0.02) were independent predictors of overall survival (OS) after accounting for Mayo stage and age. However, these markers have not been incorporated in a formal risk stratification system although they are commonly used in every day clinical practice and may guide treatment and patient management. 

#### Cardiac Imaging

##### Cardiac Echocardiography 

Cardiac echocardiography is an important tool for the diagnosis of cardiac AL and evaluation of the degree of cardiac dysfunction. Left ventricular ejection fraction measurement by echocardiography (LVEF) is typically used to assess cardiac systolic function and many studies have reported an association between low LVEF and adverse prognosis. LVEF in cardiac AL is generally preserved until late disease stages and low LVEF is therefore an indicator of very advanced cardiac amyloidosis. Kristen et al. in 2010 identified LVEF < 45% as an independent prognostic factor for OS in multivariate analysis (MVA) that included NT-proBNP and hsTnT in a prospective study of 163 newly diagnosed patients. [30] A higher cutoff value of 55% for LVEF was also reported to be an independent prognostic factor for OS in patients with newly diagnosed AL amyloidosis in two studies [5,6].

LVEF provides an estimate of the geometric changes that occur to the LV secondary to cardiac dysfunction. The quantification of longitudinal cardiac fiber function is considered to be a better measure of contractile myocardial [39]. In AL amyloidosis the subendocardial myocardium is affected first (as determined by cardiac MRI) and longitudinal fibers are mostly located in the subendocardium. This explains the prognostic implication of global longitudinal function evaluation in patients with preserved LVEF [40]. In the study by Buss et al. in 2012, in 206 consecutive patients with biopsy proven cardiac AL, echocardiography was used to assess mean tissue doppler-derived longitudinal strain (LS), and two-dimensional global longitudinal strain (2D-LGS) of the LV [32]. Reduced LS and 2D-GLS were both independently associated with OS using the cut-offs of  –10.65% and –11.78%, respectively. There was a strong correlation of both parameters with NT-proBNP. In the clinical MVA model, 2D-GLS and cTnT were independent predictors of survival in AL amyloidosis and 2D-GLS provided incremental value to the combination of NT-proBNP, cTnT, and other clinical parameters. In the study by Pun et al. in 2018 [33], in 82 patients with newly diagnosed AL amyloidosis, the cutoff value of 17% for GLS (they converted negative to positive values) was reported as the value that best discriminated survivors from non-survivors at 5 years, with an HR that was 0.91 (95% CI, 0.74–0.90, *p* < 0.001) in univariate analysis (UVA), but MVA was not performed. GLS provided added value to risk stratification within each validated cardiac staging system. Other studies have also identified GLS as a prognostic marker for outcome in AL [41,42,43]. Increased LV septum thickness as reported by echocardiography has also been linked to adverse prognosis. In two MVA models reported by Dispenzieri et al. in 2003 [5] and 2004 [6] LV septal thickness >15 mm retains its independent prognostic value, however, there are several limitations with its reproducibility. 

##### Cardiac Magnetic Resonance (MRI)

MRI is currently a necessary part of the AL amyloidosis work-up at diagnosis; it contributes to the diagnostic process and can differentiate cardiac amyloidosis from other causes of cardiomyopathy. Global diffuse myocardial late gadolinium enhancement (LGE) is more pronounced in subendocardial layers in cardiac amyloidosis and it has been associated with poor prognosis [44]. Other cardiac MRI parameters are increasingly emerging as markers of prognosis as they reflect cardiac dysfunction. Arenja et al. demonstrated in 2019 the prognostic value of longitudinal axis strain (LAS) and myocardial contraction fraction (MCF) in 74 patients with biopsy-proven AL amyloidosis [45]. LAS is a marker of longitudinal function of the LV and is defined as the percentage of longitudinal shortening of the LV between end-diastole and end-systole. MCF is calculated by dividing LV stroke volume by LV myocardial volume. Both of these parameters are unaffected by other organ dysfunction and can be extracted without dedicated processing software and without the use of gadolinium contrast which is often contraindicated in patients with renal impairment. Patients with eGFR less than 30 mL/min/1.73 m^2^ and cardiac pacemakers or other metallic implants were excluded from the study. Primary endpoint was all-cause mortality and the secondary composite endpoint included cardiac transplantation due to progressive disease. Cutoff points were determined at ≤56.6% for MCF and at <–7% for LAS. In UVA, LAS (HR = 1.05, *p* < 0.00) and MCF (HR = 0.96, *p* < 0.001) were associated with reduced transplant-free survival. Kaplan–Meier analyses for both endpoints showed reduced event-free survival in patient with LAS > –7% and MCF of 56.6%. Adding LAS and MCF to LVEF increased the predictive power of the model. Correlations between CMR-derived parameters (MCF and LAS) and cardiac NT-proBNP and cTNT was weak to moderate. 

Extracellular volume (ECV), as assessed by MRI, provides an assessment of amyloid burden in the heart, together with T1. Comparison studies have demonstrated that ECV and T1 can reliably distinguish between amyloidosis and other disease states [46]. In a study of 100 AL amyloidosis patients with a median follow up of 2 years, ECV > 44% (HR, 7.2; 95% CI, 1.751–13.179; *p* = 0.002) was an independent prognostic factor for mortality and correlated with Mayo stage and LVEF [47].

### 3.2. Markers of Renal Dysfunction 

A practical issue with the use of proteinuria in the assessment of renal impairment is that the collection of 24-h urine is inconvenient. Other alternatives have been proposed to measure the loss of protein. Other markers of renal function that have been linked to prognosis include albumin and the albumin to creatinine ratio. In univariate analysis serum albumin levels < 30 g/L predicted renal survival, however, the prognostic effect was not significant in MVA [9]. The urinary albumin to creatinine ratio (uACR) has been proposed as an alternative method to stratify and evaluate renal response to therapy and has a good correlation to 24 h urine protein. At the time of diagnosis, values of more than 500 mg/g define renal involvement while values of more than 3600 mg/g are predictive for renal survival [36]. In the course of the disease, a reduction in uACR of more than 30% is an accurate biomarker to monitor renal response to treatment, even when eGFR is below 30 mL/min/1.73 m^2^, but an increase in uACR is not as reliable for assessing renal progression [37]. Biomarkers should also be evaluated in the framework of novel therapies. In patients receiving daratumumab-based regimens, uACR seems to have an adverse effect on the efficacy of therapy-based regimens probably due to loss of monoclonal antibodies in the urine [38].

## 4. Tumor-Related Biomarkers 

Tumor-related biomarkers correspond to characteristics of the underlying clonal disorder, the burden and qualities of the clonal plasma or other B-cells, and the secreted amyloidogenic FLCs. They have emerged as biomarkers that mostly correlate with longer-term outcomes in contrast to organ-function-related markers which are more predictive of short-term outcomes. They also have significant value in response assessment, patient monitoring, and tailoring of treatment. Table 3 summarizes all tumor-related biomarkers reported to have an impact on OS to date in AL amyloidosis. 

### 4.1. Serum FLC

FLC detection and quantification provides a sensitive way to quantify the amyloid precursor protein secreted by the monoclonal plasma cell in AL amyloidosis patients who usually lack an intact pathological immunoglobulin and have small plasma cell clones compared to other plasma cell disorders [18]. It is currently considered essential for the diagnosis and monitoring of AL amyloidosis patients [2]. The introduction of a nephelometric antibody-based for the quantification of serum FLC (Freelite, Binding Site, UK) was revolutionary for the diagnosis and monitoring of patients with AL amyloidosis [20,65]. Approximately 10 years later, another assay based on monoclonal antibodies became commercially available [66]. As expected, FLC levels correlate to a degree with organ involvement and the degree of organ dysfunction as they reflect amyloidogenic protein and tumor burden. Many studies have validated the prognostic value of FLCs for OS either as the difference between the involved and uninvolved FLC (dFLC) or as the absolute value of the involved FLC (iFLC). Different dFLC and iFLC cutoffs have been reported to have prognostic value and the 180 mg/L threshold was incorporated in the revised Mayo staging system in 2012 [8,21] (Table 1). In a retrospective analysis of 93 patients with AL amyloidosis baseline iFLC (>152 mg/L) was associated with higher risk of death (HR, 2.6; *p* < 0.04), more involved organs, and higher troponin levels. iFLC was also prognostic of survival as a continuous variable (HR, 1.003; *p* = 0.013) [55]. The cutoff value of >125 mg/L retained its prognostic value for adverse OS in MVA by another group [28]. In a retrospective analysis of 783 patients dFLC of <50 mg/L was reported as an independent prognostic factor for OS in MVA associated with better outcomes (HR = 0.50, *p* = 0.003). Patients with dFLC < 50 mg/L also had lower levels of clonal plasma cells and less frequent cardiac involvement [18]. Milani et al. in 2017 [56] performed a similar study in parallel and validated the importance and the prognostic value of a low dFLC level (<50 mg/L) [56]. However, as the nephelometric FLC test cannot differentiate between clonal and polyclonal light chains in the setting of impaired renal function, the non-specific accumulation of serum FLC becomes a limiting factor in the setting of severe renal dysfunction [67].

### 4.2. Bone Marrow Plasma Cell Burden: 

Serum FLC levels are dependent on the clonal plasma cell burden [49,65] and a high number of bone marrow plasma cells (BMPC) usually translates to higher disease activity but the secretory activity of the plasma cells can be variable and sFLC levels are also dependent on the rate of elimination. Nevertheless, plasma cell burden, as assessed by bone marrow biopsy or cytology, has emerged as an adverse prognostic factor for outcome in AL amyloidosis [48,49,68].

Kourelis et al. in 2013 [48] reported that a BMPC infiltration >10% in patients with AL amyloidosis is a negative prognostic marker independent of age, Mayo stage, use of ASCT, or dFLC. A higher cutoff of ≥20% was reported by the same group in an update of the study [68] with median OS of 81, 33, and 12 months for <5%, 5–19%, and ≥20% BMPCs, respectively (*p* < 0.001). In MVA accounting for known prognostic factors, an independent prognostic role for ≥20% BMPCs but not for the other BMPC groups was demonstrated. Another group assessed circulating peripheral blood PCs (PBPCs) and reported a poorer median OS for patients with high PBPC%s (>1%) (median survival, 10 vs. 29 months, *p* = 0.002) and absolute PBPC counts (>0.5 × 10^6^)/L (median OS, 13 vs. 31 months, *p* = 0.003). Both remained independent prognostic factors for OS in MVA [54].

### 4.3. Immunophenotyping 

Multidimensional flow cytometry (MCF) has emerged as a method of high sensitivity for the detection of aberrant plasma cell clones in the BM and in the periphery. In the study by Muchtar et al. in 2017, monotypic PCs ≥ 2.5% at diagnosis, as detected by MFC was associated with shorter progression free survival (PFS) and OS compared to patients with <2.5% (2-year PFS 41% vs. 56%, *p* = 0.007; 2-year OS 55% vs. 70%, *p* = 0.01) [69]. This cutoff remained an independent predictor of adverse outcomes for both PFS/OS in MVA. MFC is prognostic for AL at diagnosis and at the time of treatment. The lower cutoff of >1% BMPC as evaluated by MFC was also reported by Paiva et al. in 2011 as an independent predictor for OS in MVA [52] (≤1% vs. >1% BMPC cutoff; 2-year OS rates of 90% vs. 44%, *p* = 0.02). 

More recently, Puig et al. [51] used an automated computerized algorithm which simultaneously assessed the degree of clonality and tumor burden and identified three patient subgroups—MGUS-like, intermediate, and MM-like. Patients with AL and a profile signature similar to MM patients had earlier mortality. The intermediate plus MM-like profiles had an independent adverse prognostic effect on PFS (HR 2.9, *p* = 0.01) and OS (HR, 3.0; *p* = 0.03). 

### 4.4. Cytogenetics of the Plasma Cell Clone 

Fluorescence in situ hybridization (FISH) abnormalities have significant prognostic value in MM and in recent years it has become apparent that the same holds true for AL. The genetic landscape is however different. There is less hyperdiploidy [70] and a much higher prevalence of translocations in the immunoglobulin heavy chain locus; translocation t(11;14) is commonly found in AL amyloidosis [63], linked to a less complex clonal landscape and the free light chain phenotype [63]. Determining the genetic profile of the plasma cell clone adds prognostic information to the assessment of the BMPC burden which is linked to more long-term outcomes and response to treatment. 

High-risk cytogenetics seen in MM (t(4;14), t(14;16), and del17) are not common in AL. More complex karyotype clones, however, and presence of del17 have an impact on outcome [64]. Gain of 1q21 is an independent adverse prognostic factor in AL patients and data from series where patients were treated with melphalan, dexamethasone, standard chemotherapy, and daratumumab [61,63]. Bochtler et al. reported an independent adverse prognostic role for 1q21 and OS in MVA (HR, 3.64; *p* = 0.003) along with established Mayo cardiac staging. 

Translocation t(11;14) is an adverse prognostic factor for patients treated with bortezomib [38,62] but this effect is overcome with daratumumab and ASCT [60,61,71]. In a group of 101 patients treated with bortezomib, the presence of t(11;14) remained an independent prognostic factor with NT-proBNP and dFLC in MVA for hematologic event-free survival (HR, 2.94; 95% CI 1.37–6.25; *p* = 0.006) and OS (HR, 3.13; 95% CI, 1.16–8.33; *p* = 0.03) [38]. In another series of 692 patients, t(11;14) was the most common abnormality seen in 49% of patients, followed by monosomy 13q in 36%. t(11;14) was associated with inferior OS in both the bortezomib (15 vs. 27 months, *p* = 0.05) and the immunomodulatory agent (IMiD)-treated groups (12 vs. 32 months, *p* = 0.05). Trisomies were also associated with poorer median OS and this effect remained significant in MVA for the entire cohort whereas the independent prognostic value of t(11;14) remained significant in MVA only in the bortezomib arm. In patients treated with HDM and ASCT the effect of t(11;14) was reversed and in a study by the same group, t(11;14) was seen in 59% of patients and had a favorable effect on outcome together with low dFLC in MVA [60]. Similarly in a cohort treated with daratumumab t(11;14) was associated with better hematological EFS, whereas gain 1q21 and hyperdiploidy were adverse factors for OS and hematological EFS [61].

### 4.5. Immunoparesis 

The role of immunoparesis as an adverse prognostic factor has also been demonstrated. In a cohort of 170 patients, using heavy light chain immunoassay (HLC) immunoparesis of at least one immunoglobulin (Ig) isotype was identified in 85% of patients and severe immunoparesis (≥2 Ig isotypes suppressed by >50% below normal levels) in 18% of patients. In patients with cardiac involvement at the 6-, 9-, and 12-month landmark analysis severe HLC suppression was associated with shorter OS (median OS 8.8 vs. 29.9 months, *p* = 0.007). A survival model which included severe HLC suppression and dFLC > 180 mg/L stratified patients into three different survival categories. The effect of immunosuppression was greatest at 6 months, at the time of chemotherapy completion. In patients with renal involvement the impact of immunoparesis requires careful assessment as nephrotic syndrome leads to IgG loss into the urine [22]. In another group of 998 patients, immunosuppression was assessed by nephelometry and defined it as either the number of suppressed uninvolved Igs below the lower limit of normal and also by assessing the average relative difference (ARD) of the uninvolved Igs from the LLN. In MVA immunosuppression as assessed by both methods retained its negative prognostic impact [57].

## 5. Prognosis and Response to Treatment 

Many of the markers with upfront prognostic value have also formed the basis of current hematologic and organ response criteria demonstrating their prognostic value in landmark analysis [72,73].

The important prognostic role of FLC levels in AL amyloidosis was confirmed by the fact that FLC assessment forms the basis of the hematologic response criteria [72]. The aim of plasma-cell directed therapy is to achieve rapid decrease and normalization of the sFLC and the quality of the FLC response is directly associated with survival and organ response [73,74]. A substantial improvement in outcomes is associated with at least a very good partial remission (a dFLC below 40 mg/L [72]. Normalization of the FLC ratio and their levels are required to establish a complete response (aCR) in addition to absence of the involved monoclonal protein component by serum and urine and negative immunofixation. [75] Given that some patients with AL amyloidosis have low baseline dFLC levels between 20 and 50 mg/L, a threshold of 10 mg/L was established as an additional hematologic response parameter (low dFLC PR) for such patients. Later this was proposed for all patients with baseline dFLC > 20 mg/L [18,76,77]. A prospective observational study has shown that this milestone translates into a benefit additional to VGPR and CR and longer time to treatment [78]. An aCR is the optimal endpoint and patients who fail to achieve it have shorter PFS. However, even a very small residual aberrant plasma cell clone can affect outcome by secreting even small amounts of toxic amyloidogenic light chains. Minimal residual disease (MRD) as assessed by next-generation flow cytometry has been used as a biomarker that can identify patients at high probability of organ response and very low probability of hematologic relapse [79,80]. 

## 6. Novel biomarkers 

### 6.1. New Prognostic Biomarkers for Survival 

A number of novel biomarkers have emerged in recent years in the attempt to improve risk stratification and outcome prediction in the AL amyloidosis population (Table 4).

#### 6.1.1. D-dimers

D-dimer levels reflect fibrin degradation and are commonly elevated in hematological malignancies. In a recent analysis of 897 patients with AL amyloidosis, 47% were found to have elevated D-dimer (>0.5μg/mL). A normal D-dimer level of ≤0.5μg/mL and a level of >0.5μg/mL but <1.0 μg/mL was associated with a lower mortality risk (HR, 0.49 and 0.59) compared to D-dimer levels ≥1μg/mL in multivariate analysis. This effect was independent of cardiac stage. The median overall survival was 5.86, 4.04, and 2.08 years for D-dimer levels of ≤0.5, >0.5 but <1, and ≥1 μg/mL, respectively (*p* < 0.001) [88]. The cause of this finding is currently unknown. It is hypothesized that it could indicate the presence of venous thromboembolism (71 patients had a clinically significant VTE) or arterial thrombi. The fibrinolytic pathway is known to be active in AL amyloidosis and could explain the increased D-dimer levels or it could be linked to underlying dysfunction and pathology in the endothelium and vasculature. The authors argue that D-dimer elevation may reflect the extent of systemic involvement in these patients. 

#### 6.1.2. Von Willebrand Factor (vWF)

In addition to the toxic effects of the circulating free light chains to the myocardium, it is hypothesized that there might be direct deposition and toxicity of the light chains to the endothelial cells (ECs) leading to altered vascular function. Von Willebrand factor (vWF) is a large multimeric glycoprotein produced, stored, and secreted mostly by the EC. vWF secretion by the EC is hypothesized to reflect EC stimulation or activation. The prognostic importance of VWF as a surrogate marker of endothelial dysfunction was assessed in 111 patients with newly diagnosed AL amyloidosis. VWF:Ag levels were significantly higher in patients with AL compared to that measured in healthy controls [87] and serum VWF:Ag ≥ 230 U/dL was associated with higher probability of early death and remained an independent predictor of death within 6 months even among patients with Mayo stage III, (1-year OS of 17% vs. 68% for patients with stage III disease and lower VWF levels, *p* < 0.001). Among patients with stage IIIb disease, high VWF levels could further discriminate them into two outcome groups (2 vs. 6 months, *p* = 0.006). 

#### 6.1.3. Red Cell Distribution Width (RDW)

A high RDW is associated with ineffective erythropoiesis, renal dysfunction, cardiovascular disease, age-related clonal hematopoiesis, and overall mortality [91,92]. In one retrospective analysis in 94 newly diagnosed patients with AL amyloidosis a high RDW was associated with adverse prognosis (a cut-off of 13.8 was determined using ROC analysis). OS was significantly shorter for patients with high RDW (*p* < 0.001), which retained its adverse prognostic power on multivariate analysis, even for Mayo stage I or patients with normal NT-proBNP [89]. 

#### 6.1.4. Soluble Suppression of Tumorigenicity 2 (sST2)

SST2 is an IL-1 receptor acting through IL-33 signalling. It is elevated in heart failure, acting as a decoy receptor of IL-33 to alleviate the cardioprotective effects of IL-33 [93]. It is hypothesized to be a marker of cardiac remodeling and myocardial fibrosis and it is predictive of mortality in patients with cardiac disease [94,95] in a manner independent of NT-proBNP and troponins [96,97]. sST2 could provide further prognostic insights for patients with AL and high troponin and NT-proBNP levels as it may indicate those who have the ability to heal/survive. It could also have implications for candidate selection for cardiac transplantation. Dispenzieri et al. [84] assessed sST2 levels and galectin-3 levels in 502 patients with AL amyloidosis. UVA patients with sST2 < 30 ng/mL vs. ≥ 30 ng/mL had significant differences in 1-year OS (81% vs. 43%) and 5-year OS (52% vs. 22%) (*p* < 0.0001). sST2 remained independent of troponin, NT-proBNP, sFLC, and blood pressure in MVA. A new scoring system incorporating ST2 was generated and a 5-level scoring system was created with a risk ratio per level of 1.8 (95% CI, 1.6–1.9). Kim et al. [82] also assessed sST2 in 73 newly diagnosed patients with AL amyloidosis and determined the optimal cut-off at 32.6 ng/mL for OS and 1 year mortality. sST2 had incremental prognostic value for OS in addition to NT-proBNP and TnT. A prognostic model including sST2 performed well even in advanced Mayo cardiac stages. 

#### 6.1.5. Osteopontin

Osteopontin (OPN) is a phosphoglycoprotein expressed and secreted by cardiomyocytes [98] among other cells and is involved in cardiac adaptation to biomechanical strain and myocardial injury. It has been linked to outcome in patients with cardiac failure and is emerging as a marker of cardiac disease severity [99,100]. In 150 patients with newly diagnosed AL amyloidosis [86], OPN levels were higher in patients with cardiac involvement and more advanced cardiac stage. The optimal cut-off was reported at 426.8 ng/mL which had independent prognostic value for all-cause mortality in MVA but it did not allow further discrimination of outcome among patients with low TnT or NT-proBNP. 

#### 6.1.6. Flow-Mediated Dilatation (FMD)

Low blood pressure in AL is most likely an indication of vascular impairment and is associated with worse outcomes. Low cardiac output, low oncotic pressure, and autonomic dysfunction have all been implicated as contributors. Another hypothesis is that low blood pressure is a compensatory mechanism. Markers of vascular reactivity could therefore add prognostic information and improve risk stratification [101]. Flow-mediated dilatation (FMD) of the brachial artery is a non-invasive marker of vascular reactivity which is augmented in conditions of hypotension and autonomic dysfunction [102,103]. Vascular reactivity in AL is relatively increased compared to controls and it is hypothesized it could correlate with autonomic dysfunction. FMD of the brachial artery was assessed in 115 newly diagnosed patients with AL amyloidosis [90]. FMD ≥ 4.5% (as determined by ROC analysis) was associated with early mortality and worse survival (hazard ratio, 2.11; 95% CI, 1.17–3.82; *p* = 0.013) even after adjustment for Mayo stage, nerve involvement and low systolic blood pressure and median OS of AL patients with FMD ≥ 4.5% was 21.3 months versus 71 months for patients with FMD < 4.5%. In MVA (model including Mayo stage and nerve involvement or for NT-proBNP, high-sensitivity troponin T or both) it remained an independent predictor of mortality at 6 months. FMD ≥ 4.5% correctly reclassified both high-risk AL patients who experienced the event and lower risk subjects who survived until the end of the follow-up over the best prognostic model (early versus late mortality).

### 6.2. New Prognostic Biomarkers for Renal Outcome 

#### 6.2.1. Growth Differentiation Factor-15 (GDF-15)

GDF-15 is a member of the transforming growth factor-β (TGF-β) cytokine superfamily produced by cardiomyocytes, macrophages, endothelial cells, vascular smooth muscle cells, and adipocytes in response to oxidative stress, inflammation, and ischemia [104]. Kastritis et al. reported that in AL amyloidosis patients’ serum GDF-15 have prognostic value for renal outcome. GDF-15 level of more than 4000 pg/mL at baseline was strong predictor for progression to dialysis (HR, 4; 95% CI, 1.16–13; *p* = 0.045) independently of the traditional renal stage. Changes in GDF-15 at 3 and 6 months post initiation of treatment were stronger predictors for renal survival than the established renal response and progression criteria (*p* =0 0.01) [81]. Importantly, a high GDF-15 level also correlated with poor OS independently of Mayo stage (HR, 1.9; 95% CI, 1.1–3.9; *p* = 0.045), while reduction of more than 25% from baseline significantly improved patients’ survival (2-year OS of 91% vs. 66%, *p* = 0.03) [81]. The results of this study were validated in an independent population.

However, GDF-15 is involved in heart failure too [105]. There was a strong correlation of GDF-15 with NT-proBNP (R^2^ = 0.211 and 0.124 in the two cohorts; *p* < 0.001 for both), hsTnT (R^2^ = 0.451 and *p* < 0.001) and Mayo stage (*p* = 0.001). There was also significant correlation of the changes in GDF-15 and NT-proBNP in the course of the disease. Another independent group has investigated the role of GDF-15 as surrogate or additive prognostic biomarker in AL amyloidosis patients. Kim et al. showed that elevated GDF-15 was associated with poor survival (sensitivity 71%, specificity 44%, *p* = 0.004 for 1-year mortality; sensitivity 80%, specificity 58%, *p* = 0.009 for overall mortality), mean wall thickness, and reduced longitudinal function of left ventricle [82]. In multivariate analysis GDF-15 was independently associated with overall mortality (HR, 2.53; 95% CI, 1.08–5.92; *p* = 0.032) [82]. 

GFD-15 is currently emerging as a marker with potential “unifying” function with prognostic value for survival, risk for dialysis, and treatment response [11]. It is elevated in >90% of all patients at diagnosis and it is a marker of poor overall outcome independent of other cardiac biomarkers (more than two thirds of patients with GDF-15 > 7575 pg/mL die within one year of diagnosis). GDF-15 levels > 4000 pg/mL are the currently the only marker of progression to dialysis independent of other biomarkers and it also performs better than the current renal staging system. GDF-15 levels are also sensitive markers of response to treatment and levels drop fast with response (greater declines compared to NT-proBNP at 3 months). Despite its seemingly promising value and role, GDF-15 needs to demonstrate reproducibility in larger series. More needs to be understood with regard to the underlying pathophysiology as currently it is unclear what the reasons are for its renal prognostic significance.

#### 6.2.2. Soluble Urokinase-Type Plasminogen Receptor (suPAR)

suPAR is the circulating form of a glycosyl-phosphatidylinositol-anchored membrane protein expressed on immunologically active cells, endothelial cells, and renal podocytes [106]. Elevated levels reflect activation of the immune system and have been associated with poor survival in various conditions such as cardiac failure and cancer [107,108]. Regarding kidney function, high suPAR levels are independently associated with chronic kidney disease (CKD) [109] and acute kidney injury (AKI) [110]. Kastritis et al. evaluated the role of suPAR as a potential novel biomarker for renal outcome. According to their results, median baseline suPAR levels were two- or three-fold higher compared to non-amyloidosis patients or the general population (6.6 vs. 3.04–3.7 vs. 2.1) [83]. In a similar way to focal segmental glomerulosclerosis, suPAR might participate in the pathogenesis of renal disease in AL amyloidosis. Higher suPAR levels at diagnosis were associated with eGFR decline (*p* = 0.008) and with renal progression as defined by Palladini criteria (*p* = 0.02). suPAR was also measured at 6 months and values over 7.2 ng/mL were strong predictors of progression to dialysis (2% vs. 20% at 2 years and 2% vs. 38% at 4 years, *p* < 0.001), independent of renal stage, renal progression, or hematologic response [83]. There was no validation cohort and their results need further investigation. 

#### 6.2.3. Galectin-3 (Gal-3)

Gal-3 is a beta-galactoside-binding lectin, involved in fibrosis and inflammation and therefore plays an essential role in the development of heart failure and kidney injury. Elevated concentrations have been associated with progressive renal impairment and with all-cause mortality in patients with non-amyloidosis renal diseases [111]. A study conducted in 2015 including AL patients, could not prove that Gal-3 had predictive value in multivariate analysis [84]. In another study by Li et al. [112], baseline serum Gal-3 levels > 20.24 ng/mL were an independent predictor of all-cause mortality, in univariate (HR, 1.92; *p* < 0.001) and multivariate analysis (HR, 2.65; *p* = 0.033). Patients were stratified into four groups based on Gal-3, hs-cTnT, and dFLC levels. Median OS was 100, 60, 29, and 15 months, respectively (*p* < 0.01) [85]. More research including patients without renal involvement is required in order to validate the prognostic value of Gal-3 and whether this stratification system can be applied to the management of systemic AL amyloidosis. 

## 7. Conclusions

Outcome in patients with AL amyloidosis is dependent on the synergistic effects of the baseline pattern and severity of organ involvement, the underlying clonal disease burden and biology, and the effectiveness of response to treatment. Given the detrimental impact of cardiac involvement on outcome, overall survival is mostly determined by cardiac involvement. Biomarkers of organ function are associated more with early survival and mortality. European modification of Mayo cardiac staging performs best. Cardiac staging is used in the clinical trial design setting to risk-stratify patients and tailor treatment. Renal biomarkers predict risk of progression to dialysis and renal staging systems have not yet currently been incorporated into clinical practice. Factors that reflect and characterize the underlying tumor disease burden and biology determine longer-term outcomes and responses to treatment. sFLC level and minimal residual disease assessment (by NGF) form the current and future basis of hematologic response criteria. Staging systems are continuously evolving and changing as are response criteria. Novel biomarkers have emerged in recent years and their value remains to be validated. A biomarker with a “unifying” role in terms of prognostic value for survival, risk of dialysis, and treatment response has not yet been identified. It is, however, unlikely that a single biomarker will be able to reflect the complexity and heterogeneity of the disease. It should, of course, be noted that current outcomes are determined by diagnostic tools and available therapeutic options which therefore indirectly affect the performance of staging systems and individual biomarkers. As new treatments emerge, current risk-stratification tools are expected to change and adapt in terms of applicability and value. 

## Figures and Tables

**Table 1 ijms-22-10916-t001:** Mayo staging systems and renal staging systems. NT-proBNP, amino-terminal portion of pro-brain natriuretic peptide type B; BNP, natriuretic peptide type B; cTnT, cardiac troponin T; cTnI, cardiac troponin I; hs-cTnT, high-sensitivity cardiac troponin; dFLC, difference between involved and uninvolved free light chain concentration; eGFR, estimated glomerular filtration rate.

	Markers and Cutoffs	Stages	Median OS, Months	HR for OS
Cardiac				
Mayo 2004 [6]	NT-proBNP > 332 ng/L BNP > 81 ng/L cTnT > 0.035 ng/mL (cTnI > 0.01 ng/mL)	I: no marker above the cutoff	130	Reference
		II: one marker above the cutoff	54	2.3
		III: both markers above the cutoff	10	6.4
European modification [7]	Like Mayo 2004 Mayo III is divided into two groups NT-proBNP > 8500 ng/L (or BNP 700 ng/L)	I	130	Reference
		II	54	2.4
		IIIa: both markers above the cutoff and NT-proBNP < 8500 ng/L	24	4.2
		IIIb: Mayo stage III and NT-proBNP > 8500 ng/L	4	11.3
Cardiac + Tumor-related				
Mayo 2012 [8]	NT-proBNP > 1800 ng/L	I: no marker above cutoff	130	Reference
	cTnT > 0.25 ng/mL	II: 1 marker above cutoff	72	1.8
	dFLC > 180 mg/L	III: 2 markers above cutoff	24	3.7
		IV: 3 markers above cutoff	6	7.1
Renal			Risk of Dialysis	
Palladini et al. 2014 [9]	eGFR < 50 mL/min/1.73 m^2^	I: both eGFR and proteinuria below cutoff	0% risk of dialysis at 3 years	
	proteinuria > 5 g/24 h	II: either eGFR below or proteinuria above the cutoffs	7% risk of dialysis at 2 years	
		III: both eGFR below and proteinuria above the cutoff	60% risk of dialysis at 2 years	

**Table 2 ijms-22-10916-t002:** Prognostic significance of organ-function related biomarkers in AL amyloidosis. NT-proBNP, amino-terminal portion of pro-brain natriuretic peptide type B; MVA, multivariate analysis; UVA, univariate analysis; OS, overall survival; EFS, event-free survival; RS, renal survival; BNP, natriuretic peptide type B; cTnT, cardiac troponin T; cTnI, cardiac troponin I; hs-cTnT, high-sensitivity cardiac troponin; LV, left ventricle; NYHA, New York Heart Association; echo, echocardiogram; ECF, electrocardiogram; MRI, cardiac magnetic resonance tomography; LVEF, left ventricular ejection fraction; GLS, global longitudinal strain; eGFR, estimated glomerular filtration rate.

Biomarkers		Thresholds	Prognostic Significance	Reference Number
NT-proBNP	serum	>152/>332/>18,000/>2736 pg/ML >8500 pg/ml	Adverse OS in MVA Adverse OS in MVA	[6,7,8,28]
BNP	Serum	>81 pg/mL >700 pg/mL	Equivalent to NT-proBNP > 332 pg/mL Equivalent to NT-proBNP > 8500 pg/mL	[29]
cTnT	Plasma Serum	Continuous or >0.03/≥0.035 μg/L	Adverse OS in MVA	[5,6,8]
cTnI	Plasma Serum	Continuous or >70/>100 ng/L	Adverse OS in MVA	[5,6,28]
hs-cTnT	Plasma serum	Continuous or ≥14/5054/ >77 ng/L	Adverse OS in MVA	[28,30,31]
Ejection fraction	Echo (LVEF)	EF continuous or <45%/50%	Adverse OS in MVA	[5,6,30]
LV longitudinal function	Echo (GLS)	GLS < −11.8% GLS 17%	Adverse OS in MVA discriminated survivors from non-survivors and added prognostic value within each cardiac staging system	[32,33]
LV septum thickness	Echo	>15 mm	Adverse OS in MVA	[5,6]
NYHA class		>2	Adverse OS in MVA	[28]
Atrial arrhythmia	ECG	Presence	Adverse OS in MVA	[25,34]
Systolic blood pressure		<100 mmHg	Adverse OS in MVA	[7]
Uric acid	Serum	>8 mg/dL	Adverse OS in MVA	[35]
Albumin-to creatinine ratio		3600 mg/gr 3600 mg/gr 220 mg/mmol	Adverse RS Adverse RS Adverse EFS	[36,37,38]
Albumin		cont or ≤30 g/L		[6,9]
Proteinuria		>5 g/24 h		[6,9]
EGFR		<50 mL/min	Adverse OS in MVA Adverse RS in MVA	[9,25]

**Table 3 ijms-22-10916-t003:** Prognostic significance of tumor-related biomarkers. BM, bone marrow; MVA, multivariate analysis; UVA, univariate analysis; OS, overall survival; PFS, progression-free survival; EFS, event-free survival; MCF, multiparameter flow cytometry; PC, plasma cells; BMPC, bone marrow plasma cells, CBPCs, circulating blood plasma cells; M protein, monoclonal protein; FLC, free light chain; iFLC, involved free light chain; dFLC, difference between involved and uninvolved FLC; HLC, heavy light chain; LLN, lower limit of normal; FISH, fluorescence in situ hybridization; HDM, high-dose melphalan in autologous stem cell transplant; M-dex, melphalan dexamethasone.

Factor/Biomarker		Thresholds or Adverse Factors	Prognostic Signfiicance	Reference
Bone marrow	BM cytology or histology	≥10%	MVA adverse OS and PFS	Kourelis 2013 [48] Tovar 2018 [49]
		≥20%	MVA adverse OS	Muchtar 2019 [50]
	BM-MCF	>1%, ≥2.5%, MFC automated profile	MVA adverse OS and PFS	Puig 2019 [51]
	BM MCF	>1% clonal PC	MVA adverse OS and PFS	Paiva 2011 [52]
		≥2.5% clonal PC	MVA adverse OS and PFS	Muchtar 2017 [53]
	CBPCs	> 5 × 10^6^/L or >1%	Adverse OS (limited MVA)	Pardanani 2003 [54]
M protein	Urine	Continuous or >1 g/24 h	MVA adverse OS	Dispenzieri 2003 [5] Dispenzieri 2004 [6]
FLC	Serum iFLC	>125 mg/L	MVA adverse OS	Palladini 2010 [28]
		>152 mg/L and continuous	MVA adverse OS	Dispenzieri 2006 [55]
	dFLC	> 50 mg/L	MVA adverse OS, more frequent and severe heart involvement	Dittrich 2017 [18] Milani 2017 [56]
		>196 mg/L, λ > 182/κ > 294 mg/L	MVA for OS and more frequent and severe heart involvement	Kumar 2010 [21]
		>180 mg/L	MVA for OS	Kumar et al. 2012 [8] J Sachchithanantham 2017 [22]
Immuno-paresis	HLC immunoassay	Severe HLC immunosuppression (≥2 Ig isotypes suppressed by >50% below normal levels)	Cardiac involvment in landmark analysis at 6 months MVA OS	Sachchithanantham 2017 [22]
	Ig nephelometry	Number of IgGs under the LLN and average relative difference of uninvolved IgGs from the LLN	MVA in OS	Muchtar 2017 [57]
Any chromosomal aberration	FISH/BM	Presence	Increased plasmacytosis, cardiac involvment, adverse OS in MVA	Hammons 2018 [58]
			Adverse OS in MVA, cardiac involvement	Warsame 2015 [59]
t(11;14)	FISH/BM	Presence	Adverse OS when BMPC ≤ 10%	Warsame 2015 [59]
			Favorable OS after HDM/daratumumab	Bochtler 2016 [60] Kimmich 2020 [61]
			Adverse OS in MVA	Bochlter 2015 [38]
			Adverse OS in MVA for bortezomib based regimens	Muchtar 2017 [62]
Gain of 1q21	FISH/BM	Presence	Adverse OS and EFS in MVA for M-dex	Bochtler 2014 [63]
			Adverse OS and EFS in UVA for daratumumab	Kimmich 2020 [61]
Deletion of 17p	FISH/BM	>50% cells	Trend towards short OS	Wong 2018 [64]
Trisomies	FISH/BM	presence	Adverse OS in MVA	Muchtar 2017 [62]
			Adverse OS when BMPC > 10%	Warsame 2015 [59]

**Table 4 ijms-22-10916-t004:** Prognostic significance of novel biomarkers in AL amyloidosis. GDF-15, growth differentiation factor-15; suPAR, soluble urokinase-type plasminogen receptor; Gal-3, galectin-3; vWF, von Willebrand factor; RDW, red cell distribution width; sST2, soluble suppression of tumorigenicity 2; LV, left ventricle; ECV, extracellular volume; FMD, flow-mediated dilatation; MRI, magnetic resonance tomography; MCF, myocardial contraction fraction of the LV; LAS, longitudinal axis strain; RS, renal survival; MVA, multivariate analysis; OS, overall survival; UVA, univariate analysis.

Biomarkers		Thresholds	Prognostic Significance	Reference Number
GDF-15	Serum	4000 pg/mL 7575 pg/mL 2300 pg/mL	Adverse RS MVA and Adverse OS MVA Adverse OS MVA	[81] [81] [82]
suPAR	Serum	7.2 at 6 months	Adverse RS	[83]
Gal-3	Serum	11 ng/mL 20.24 ng/mL	Adverse OS in UVA Adverse OS MVA	[84] [85]
Osteopontin	Serum	>426 ng/mL	Adverse OS in MVA no significant predictive value	[86] [82]
vWF	Serum	≥230.0 U/dL	Adverse OS MVA	[87]
D-dimer	Serum	D-dimer ≥ 1 μg/mL vs. <0.5μg/mL and vs. <1.0 but >0.5 μg/mL	Increased risk of mortality in MVA	[88]
RDW	Serum	RDW ≥ 13.8	Adverse OS MVA Also in subgroup with no cardiac involvement	[89]
sST2	Serum	>32.6 ng/mL ≥30 ng/mL	Adverse OS in MVA and 1-year survival Adverse OS in MVA	[82] [84]
Myocardial contraction fraction	MRI	MCF ≤ 56.6%	Adverse OS in MVA	[45]
LV longitudinal axis strain	MRI	LAS < −7%	Adverse OS in MVA	[45]
ECV	MRI	>0.45	Higher mortality	[47]
FMD	Doppler	≥4.5%	Adverse OS in MVA	[90]

## Data Availability

Not applicable.

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
