# Peer review of "Biomarkers in AL Amyloidosis"

_ijms, 2021, doi:10.3390/ijms222010916_

Round 1

Reviewer 1 Report

Fotiou et al reviewed the currently used biomarkers in AL amyloidosis and the unmet needs. Thank you, this paper addressed many questions especially in imaging biomarkers and AL. Just a one minor comment below;
- If you already explained the abbreviation in the first place, you do not need to give the whole word again. (such as Cardiac magnetic resonance (MRI) - Cardiac magnetic resonance imaging is currently...)

Author Response

Thank you very much for your comments. We have made the particular correction and we have reviewed all the text to make sure all the abbreviations are used correctly and consistently throughout the text. 

Reviewer 2 Report

With pleasure I read the review “Biomarkers in AL Amyloidosis” by Fotiou et al. The content reflects very well the current state of the literature. The article is well written and in my opinion publishable in IJMS after minor improvements.

Regarding the content:

Line 345:

“but this effect is overcome with daratumumab and ASCT. [64,67]”.

I think one could also cite the Andomeda Study here.

Often the abbreviations are not introduced, the whole manuscript should be reviewed in this regard. Specifically, I noticed:

NT-proBNP is introduced several times and differently in the course of the manuscript. I think N-terminal pro Brain natriuretic peptide is the most common term.

Line 125: ESRD needs to be introduced.

Line 193: MVA needs to be introduced.

Line 174, but also other times in the course: echo is colloquialism in my opinion. It should be echocardiography.

Author Response

Thank you very much for all your comments and input.

  1. Often the abbreviations are not introduced, the whole manuscript should be reviewed in this regard. We have reviewed the manuscript and have introduced all the abbreviations appropriately.
  2. Specifically, I noticed:

NT-proBNP is introduced several times and differently in the course of the manuscript. I think N-terminal pro Brain natriuretic peptide is the most common term. We have used the term N-terminal pro Brain natriuretic peptide to introduce the abbreviation and we have used it consistently throughout the manuscript. 

  1. Line 345: “but this effect is overcome with daratumumab and ASCT. [64,67]”. I think one could also cite the Andomeda Study here: We have  added the relevant reference for the Andromeda Study 
  2. Line 125: ESRD needs to be introduced. We have introduced the term: End stage renal disease

  3. Line 193: MVA needs to be introduced. We have introduced the term Multivariate analysis 

  4.  

    Line 174, but also other times in the course: echo is colloquialism in my opinion. It should be echocardiography. We have made all the relevant changes and have used the word echocardiography instead of echo.